# PLuG-Attention: Unleashing the Potential of Attention via Plug-in Pairwise Logit Gating

## Abstract

Despite its widespread success on vision tasks, standard attention employs a shared dot-product mechanism that uniformly scores all query–key interactions before applying softmax. In this paper, we hypothesize that explicitly controlling the amplification or suppression of individual query-key token-pair interactions can lead to more expressive and discriminative representations. To this end, we propose **Pairwise Logit Gating (PLuG)** attention, a simple yet effective plug-in approach that introduces a learnable gating mechanism operating on each token-pair to modulate attention logits prior to softmax. This gating enables the model to selectively amplify informative interactions and suppress spurious ones through gating coefficient matrix, improving its ability to capture spatial and semantic relationships critical for vision tasks. Experimental results demonstrate that PLuG can be seamlessly integrated into various attention mechanisms and attention-based architectures, including ViTs and Mask2Former, as well as multi-scale deformable attention in Deformable DETR, without requiring architectural redesign or hyperparameter tuning. These results highlight the effectiveness of PLuG as a general-purpose plug-in enhancement broadly applicable to attention-based vision tasks.

## 1 Introduction

Since the Transformer's introduction (Vaswani et al., 2017), attention mechanisms have established themselves as a foundational component of deep learning, enabling advances in natural language processing (Devlin et al., 2019; Radford et al., 2018). The self-attention mechanism computes pairwise token interactions via dot-product similarity between query and key representations, followed by softmax normalization to produce attention weights that modulate the aggregation of value features. This design enables the modeling of global context and long-range dependencies, addressing the locality limitations inherent in convolutional architectures with fixed receptive fields. Based on this approach, a wide range of vision architectures such as Vision Transformers (ViT) (Dosovitskiy et al., 2021) for image classification, DETR (Carion et al., 2020) for object detection, and Mask2Former (Cheng et al., 2022) for semantic segmentation have successfully extended attention-based models to diverse visual recognition tasks.

Despite their remarkable success, standard attention mechanisms employ a shared dot-product mechanism that uniformly scores all query–key interactions before softmax, regardless of their semantic or spatial relevance. This uniform scoring can lead the model to focus on background regions or non-informative structures while overlooking salient visual cues. As a result, the lack of dynamic modulation at the token-pair level may limit the model's ability to adapt its focus based on image content, reducing its effectiveness on vision tasks that require fine-grained spatial understanding and semantic awareness.

In this paper, we hypothesize that explicitly controlling the amplification or suppression of individual token-pair interactions can lead to more expressive and discriminative representations. To this end, we propose **Pairwise Logit Gating (PLuG)** attention, a simple yet effective plug-in mechanism that introduces learnable, token-pair-specific gating mechanisms to modulate attention logits prior to softmax normalization. Specifically, PLuG computes auxiliary query-key projections using linear layers with dimensionality equal to the per-head dimension. These auxiliary queries and keys are then projected into a shared space and compared using scaled dot-product similarity, yielding a raw

pairwise gating matrix. This matrix captures token-pair interactions in a low-dimensional gating space. To transform this raw matrix into meaningful modulation signals, we introduce a Gating Modulation Layer (GML), a lightweight two-factor scheme that maps the raw gating matrix to two learned components through a small linear layer. These components are then multiplied elementwise and passed through a tanh activation to produce a bounded gating coefficient matrix, which plays a crucial role in improving the expressiveness of attention mechanisms. Finally, as illustrated in Figure 1(a), this gating coefficient matrix is applied multiplicatively to the standard attention logits before softmax.

Figure 1(b) demonstrates that PLuG enables fine-grained control by learning to modulate attention logits at the token-pair level in DeiT-S (Touvron et al., 2021a), suppressing self and local interactions while amplifying medium and long-range dependencies. This structured modulation helps the model reduce reliance on uninformative or spurious correlations, guiding attention toward more semantically and spatially meaningful dependencies. Notably, PLuG integrates seamlessly into ViT-based architectures and Mask2Former by modifying only the attention module, and it also applies to multi-scale deformable attention in Deformable DETR (Zhu et al., 2020), without altering the rest of the architecture. Importantly, this improvement is achieved with only a small increase in computational cost and works effectively without hyperparameter tuning, making PLuG a practical, plug-in solution for attention-based vision models. The contributions of this paper are summarized as follows:

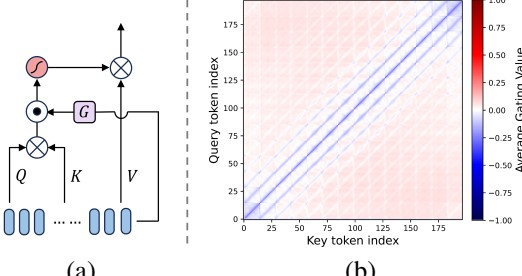

(a)                    (b)

Figure 1: (a) Overview of PLuG Attention. (b) Visualization of the average gating coefficient matrix for DeiT-S (Touvron et al., 2021a) with PLuG applied, computed over all heads and layers on ImageNet-1k (Russakovsky et al., 2015) val images. Blue indicates suppression, while red signifies amplification, reflecting the modulation applied to the pre-softmax attention logits between query–key pairs.

- We propose PLuG attention, which modulates attention logits prior to softmax through learnable token-pair-specific gating mechanisms, enabling selective amplification of informative interactions while suppressing spurious ones.

- We demonstrate that PLuG can be seamlessly integrated into various attention mechanisms and attention-based architectures, including ViTs, Mask2Former, and multi-scale deformable attention in Deformable DETR. To the best of our knowledge, this is the first work to introduce and validate a logit-level gating mechanism consistently across domains, architectures, and attention variants in vision tasks.

- Extensive experiments show that PLuG is a true plug-in, applied only within within the attention module, requiring no architectural redesign or hyperparameter tuning, and serving as a practical, general-purpose enhancement for attention-based vision models.

## 2  RELATED WORKS

**Token-Pair Modulations**    Several recent approaches alter how token pairs interact. Talking-Heads Attention (Shazeer et al., 2020) learns small projections that remix pre-softmax logits across heads, providing pairwise modulation. Synthesizer (Tay et al., 2021) replaces dot-products by generating full attention maps with tokenwise MLPs or global random weights, removing explicit query–key similarity while retaining strong performance. ConViT (d'Ascoli et al., 2021) instead mixes post-softmax content and positional distributions with a per-head sigmoid gate and then renormalizes, applying the same scalar mixture weight to all token pairs within a head rather than rescaling logits directly. Selective Self-Attention (Zhang et al., 2024a) applies a query-specific temperature to an entire row, not individual pairs. Another method, also titled Selective Attention (Leviathan et al., 2025), derives a parameter-free, non-negative mask from one of the existing attention heads and subtracts it from the logits, thereby suppressing specific token pairs but not amplifying them. Compared with these, PLuG applies lightweight, learnable gates directly on pre-softmax logits at

the token-pair level, allowing it to both amplify and suppress specific pairs for precise, fine-grained modulation.

**Attentions using Gating Mechanisms**  Another line of research integrates gating mechanisms after attention weight computations. For instance, Gated Attention Unit (GAU) (Hua et al., 2022) and MEGA (Ma et al., 2023) apply learned gates to token representations after the attention operation, achieving improved efficiency in long-range modeling. Gated Linear Attention (GLA) (Yang et al., 2024) introduces data-dependent forget gates within linear-time Transformers, enabling scalability to extended sequences and high-resolution image generation tasks such as (Zhu et al., 2025). Gated Slot Attention (GSA) (Zhang et al., 2024b) further augments bounded-memory attention with a gating mechanism while retaining softmax, improving recall-intensive performance and training/inference efficiency. Task-specific architectures also apply gating at different stages of the attention process. AVIGATE (Jeong et al., 2025) applies learned scalar gates to the outputs of cross-attention and FFN within a gated fusion transformer, modulating cross-modal features after attention rather than altering pre-softmax logits. The Gated Attention Transformer (Doering & Gall, 2023) fuses appearance and pose similarity at the logit level via an $\alpha - Gate$, forming a weighted sum of pairwise scores before softmax. Unlike prior methods, PLuG requires no architectural changes and hyperparameter tuning. We further show that it integrates seamlessly into diverse attention mechanisms and architectures, including ViTs, Mask2Former, and multi-scale deformable attention in Deformable DETR, achieving strong results across image classification, detection, and segmentation. To the best of our knowledge, this is the first logit-level gating mechanism validated consistently across domains, architectures, and attention variants in vision tasks. Further discussion is provided in Appendix F.

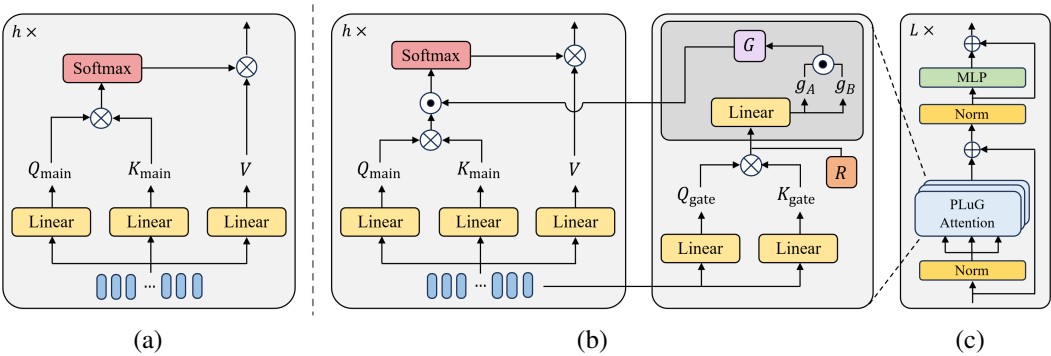

(a)  (b)  (c)

Figure 2: Illustration of PLuG Attention Mechanism. (a): Standard multi-head attention computes attention weights from the similarity between $Q_{\mathrm{main}}$ and $K_{\mathrm{main}}$, followed by softmax normalization. (b): PLuG attention introduces learnable gating matrix $G$ that modulate attention logits prior to softmax. The gates are computed from separate $Q_{\mathrm{gate}}$ and $K_{\mathrm{gate}}$ projections, enabling selective amplification or suppression of token-pair interactions. Darker region highlights the GML process, which produces the gating coefficient matrix $G$. (c): PLuG is applied by modifying only the attention module, without altering the rest of the architecture.

## 3 METHODOLOGY

### 3.1 PLuG ATTENTION

We introduce Pairwise Logit Gating (PLuG) Attention, enabling dynamic modulation of the token-pair interaction weights based on learned gating functions. PLuG operates as a true plug-in enhancement, requiring only changes to multi-head self-attention (MHSA) layer of existing attention architectures. The overall architecture is illustrated in Figure 2.

**Main Attention Path**  In a standard self-attention block of a ViT (Dosovitskiy et al., 2021), an input image of spatial resolution $(H, W)$ is divided into non-overlapping patches of size $(P, P)$. Each patch is flattened to form a token sequence $x \in \mathbb{R}^{N \times C}$, where $N = HW/P^2 + 1$ includes the class token. The sequence is linearly projected into queries, keys, and values:

$$Q_{\mathrm{main}} = xW_q, \qquad K_{\mathrm{main}} = xW_k, \qquad V = xW_v, \tag{1}$$

where $W_q, W_k, W_v \in \mathbb{R}^{C \times C_v}$ and $C_v = h\,d_h$ denotes the projection dimension, which is typically equal to $C$. Scaled dot-product attention logits are then computed as

$$A_{\text{main}} = \frac{Q_{\text{main}} K_{\text{main}}^\top}{\sqrt{d_h}}, \tag{2}$$

with $h$ denoting the number of heads and $d_h$ the per-head dimension. Each head attends independently, and the outputs are concatenated to dimension $C_v$ before being projected back to the embedding space.

**Gating Path** To explicitly modulate each token-pair interaction, we introduce a gating mechanism operating in parallel to the main attention path. The gating mechanism begins by computing gating-specific queries $Q_{\text{gate}}$ and keys $K_{\text{gate}}$:

$$Q_{\text{gate}} = x W_{q_g}, K_{\text{gate}} = x W_{k_g}, \tag{3}$$

where $W_{q_g}, W_{k_g} \in \mathbb{R}^{C \times d_g}$. $Q_{\text{gate}}$ and $K_{\text{gate}}$ are shared across heads and form the raw pairwise gating matrix $R \in \mathbb{R}^{N \times N}$ via a scaled dot-product:

$$R = \frac{Q_{\text{gate}} \cdot K_{\text{gate}}^\top}{\sqrt{d_g}}, \tag{4}$$

This raw gating matrix $R$ captures token-pair interactions in a low-dimensional gating space. Here, the gating dimension $d_g$ is chosen as a fraction of the head dimension $d_h$. The effect of different gating dimensions is illustrated in Figure 4.

**Gating Modulation Layer** Subsequently, we introduce a simple nonlinear gating module, Gating Modulation Layer (GML), that maps the raw gating matrix $R$ into a flexible modulated gating coefficient matrix $G$. Specifically, each scalar value in raw gating matrix $r_{ij} \in R$ is transformed by a lightweight linear layer parameterized by $W_\phi \in \mathbb{R}^{1 \times 2}$, yielding two scalar modulation factors denoted as $g_A, g_B \in \mathbb{R}^{N \times N}$. For clarity, the bias term is omitted from the notation but included in implementation. The two factors are then multiplied elementwise and passed through a hyperbolic tangent activation to produce the gating coefficient matrix $G = \tanh(g_A \odot g_B) \in \mathbb{R}^{N \times N}$, where $\odot$ denotes elementwise multiplication. The tanh activation stabilizes training by preventing unbounded logits that can lead to loss divergence, while the resulting gating matrix $G$ enables dynamic, token-pair-specific modulation of attention that enhances model performance. The full process of GML is illustrated in Figure 3.

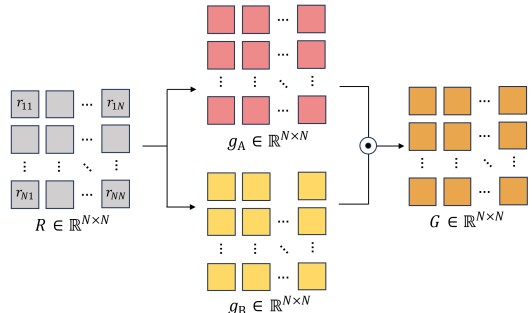

Figure 3: Illustration of the Gating Modulation Layer (GML). GML maps pairwise raw gating matrix $R$ into a gating coefficient matrix $G$, which plays a critical role in enhancing model performance by dynamically modulating attention logits in a token-pair-specific manner.

**Attention Modulation** The final attention logits are obtained by multiplicatively modulating the main attention logits $A_{\text{main}}$ with the learned gating coefficients $G$:

$$\tilde{A}_{\text{main}} = A_{\text{main}} \odot (1 + G), \tag{5}$$

where $G \in \mathbb{R}^{N \times N}$ operates at the token-pair level and is broadcast across all heads in ViT-based settings (Dosovitskiy et al., 2021). The residual term "1+" preserves the original attention logits, allowing the model to flexibly amplify or suppress them as needed. This modulation explicitly increases attention on informative pairs ($G > 0$) and decreases it on less relevant ones ($G < 0$). Finally, $\tilde{A}_{\text{main}}$ is normalized by softmax and the output is computed as in standard MHSA:

$$\text{head}_h = \tilde{A}_{\text{main}}^{(h)} V^{(h)}, \tag{6}$$

$$y = \text{Concat}(\text{head}_1, ..., \text{head}_H) W_O, \tag{7}$$

followed by linear projection $W_O \in \mathbb{R}^{C_v \times C}$ and dropout layers (Srivastava et al., 2014).

### 3.2 PLuG Attention for Multi-Scale Deformable Attention

Unlike ViT (Dosovitskiy et al., 2021), which models pairwise query–key interactions across a dense attention matrix, multi-scale deformable attention (MSDeformAttn) (Zhu et al., 2020) computes attention by sampling a fixed set of points for each head and feature level, rather than evaluating all query–key pairs. To match this design, we employ a head-specific, level-specific, point-shared gate that modulates logits at the $(h, \ell)$ granularity and is broadcast across sampling points. This avoids a dense pairwise gate while preserving efficiency and demonstrates PLuG's applicability beyond MHSA (Vaswani et al., 2017).

Specifically, given a query embedding $z_q \in \mathbb{R}^C$, we first compute the unnormalized attention logits as

$$A_{\text{main}} = z_q W_q, \quad W_q \in \mathbb{R}^{C \times C_v}, \quad C_v = H \times L \times P, \tag{8}$$

and reshape $A_{\text{main}}$ to $\mathbb{R}^{H \times L \times P}$. Here $H$, $L$, and $P$ denote the number of heads, feature levels, and sampling points per head–level pair, respectively. Next, we compute raw gate logits for each head–level combination:

$$r = z_q W_g, \quad W_g \in \mathbb{R}^{C \times C_v'}, \quad C_v' = H \times L, \tag{9}$$

and reshape $r$ to $\mathbb{R}^{H \times L \times 1}$. Each scalar $r_{h,\ell}$ is passed through a lightweight linear map $W_\phi \in \mathbb{R}^{1 \times 2}$ (bias omitted in notation but included in implementation), producing two factors $g_{\text{A}}, g_{\text{B}} \in \mathbb{R}^{H \times L \times 1}$. We then define the gate via:

$$G = g_{\text{A}} \odot g_{\text{B}} \in \mathbb{R}^{H \times L \times 1}. \tag{10}$$

Because $A_{\text{main}}$ has an additional sampling-point dimension, the same $G$ is broadcast along $P$:

$$\tilde{G}_{h,\ell,p} = G_{h,\ell,1}, \qquad p = 1, \dots, P, \tag{11}$$

giving $\tilde{G} \in \mathbb{R}^{H \times L \times P}$. We then apply a residual multiplicative modulation to the logits as

$$\tilde{A}_{\text{main}} = A_{\text{main}} \odot (1 + \tilde{G}), \tag{12}$$

followed by a softmax over the $L \times P$ elements within each head. Unlike the ViT-based PLuG mechanism, no additional nonlinearity is applied to $\tilde{G}$ in the MSDeformAttn setting. All remaining operations follow (Zhu et al., 2020).

## 4 Experiments

We apply PLuG to various attention-based architectures by modifying only the attention modules. Unless specified, all other components, including backbone design, training settings, loss functions, and hyperparameters remain unchanged. All models were trained from scratch. Detailed information for producing our results are provided in Appendix C.

### 4.1 Results

**PLuG to ViT-based Architectures**   We evaluate PLuG on a range of ViT-based (Dosovitskiy et al., 2021) architectures with diverse design characteristics, as summarized in Table 1. On DeiT-Ti and DeiT-S (Touvron et al., 2021a), which rely entirely on global self-attention without convolutional locality, PLuG reaches 73.2% and 80.3%, showing strong benefits in pure Transformer baselines. TinyViT-5M (Wu et al., 2022), which incorporates window attention and depthwise convolutions (Chollet, 2017), attains 79.7%. Visformer-Ti (Chen et al., 2021), with early convolutional blocks before global attention, achieves 78.5%, confirming that PLuG provides added value even in hybrids that already encode locality. LV-ViT-T (Yuan et al., 2021), equipped with a convolutional stem and dense token-label supervision, improves to 79.4%, while XCiT-T12/16 (El-Nouby et al., 2021), which reformulates attention as cross-covariance along channels, reaches 77.4%. For PVT-Tiny (Wang et al., 2021), which employs spatial-reduction attention, PLuG yields 75.2%. GCViT-XXT (Hatamizadeh et al., 2023), alternating local and global attention within fused MBConv layers, gains substantially, reaching 80.5%. ViTAE-T (Xu et al., 2021), which augments Transformers with convolutional reduction and normal cells to inject locality and multi-scale context, similarly reaches 75.7%. T2T-ViT-10 (Yuan et al., 2021), which introduces a tokens-to-token module for

Table 1: Results of applying PLuG to various ViT-based models on ImageNet-1k (Russakovsky et al., 2015) classification.

| Model | Top-1 Acc (%) | #Params | FLOPs |
|---|---|---|---|
| DeiT-Ti (Touvron et al., 2021a) | 72.2 | 5.7M | 1.3G |
| + PLuG | **73.2** | 6.0M | 1.4G |
| DeiT-S (Touvron et al., 2021a) | 79.8 | 22.0M | 4.6G |
| + PLuG | **80.3** | 22.6M | 4.9G |
| TinyViT-5M (Wu et al., 2022) | 79.1 | 5.3M | 1.3G |
| + PLuG | **79.7** | 5.5M | 1.4G |
| Visformer-Ti (Chen et al., 2021) | 78.1 | 10.3M | 1.3G |
| + PLuG | **78.5** | 10.8M | 1.3G |
| LV-ViT-T (Yuan et al., 2021) | 79.1 | 8.5M | 2.9G |
| + PLuG | **79.4** | 8.8M | 3.1G |
| XCiT-T12/16 (El-Nouby et al., 2021) | 77.1 | 6.7M | 1.2G |
| + PLuG | **77.4** | 6.9M | 1.3G |
| PVT-Tiny (Wang et al., 2021) | 75.1 | 13.2M | 1.9G |
| + PLuG | **75.2** | 13.3M | 2.0G |
| GCViT-XXT (Hatamizadeh et al., 2023) | 79.9 | 12.0M | 2.1G |
| + PLuG | **80.5** | 12.2M | 2.3G |
| ViTAE-T (Xu et al., 2021) | 75.3 | 4.8M | 1.5G |
| + PLuG | **75.7** | 5.1M | 1.6G |
| T2T-ViT-10 (Yuan et al., 2021) | 75.2 | 5.9M | 1.5G |
| + PLuG | **75.3** | 6.2M | 1.6G |

progressive structural modeling, also shows an improvement, although the gain is modest. Overall, PLuG improves accuracy across backbones, and the scale of the gains and the related increases in parameters and FLOPs depend on the underlying architectural design.

Table 2: Results of applying PLuG to Mask2Former (Cheng et al., 2022) across different backbones on ADE20K (Zhou et al., 2019).

| Model | Backbone | Crop Size | mIoU | #Params | FLOPs |
|---|---|---|---|---|---|
| Mask2Former (Cheng et al., 2022) | R50 | $512 \times 512$ | 47.2 | 44.0M | 70.8G |
| + PLuG | R50 | $512 \times 512$ | **47.9** | 44.1M | 70.8G |
| Mask2Former (Cheng et al., 2022) | Swin-T | $512 \times 512$ | 47.7 | 47.4M | 73.7G |
| + PLuG | Swin-T | $512 \times 512$ | **48.7** | 47.6M | 73.7G |
| Mask2Former (Cheng et al., 2022) | Swin-S | $512 \times 512$ | 51.3 | 68.8M | 97.4G |
| + PLuG | Swin-S | $512 \times 512$ | **51.7** | 68.9M | 97.5G |
| Mask2Former (Cheng et al., 2022) | Swin-B | $640 \times 640$ | 52.4 | 106.9M | 223.4G |
| + PLuG | Swin-B | $640 \times 640$ | **52.6** | 107.1M | 223.5G |

**PLuG to Mask2Former**  For semantic segmentation on ADE20K (Zhou et al., 2019), PLuG consistently improves Mask2Former (Cheng et al., 2022) across different backbones, as shown in Table 2. With ResNet-50 (He et al., 2016), PLuG achieves 47.9 mIoU, while Swin-T, Swin-S, and Swin-B (Liu et al., 2021) reach 48.7, 51.7, and 52.6, respectively. Despite adding only negligible computational overhead, these improvements demonstrate the effectiveness of PLuG as a plug-in module for dense prediction tasks, further underscoring its generality across both recognition and segmentation.

**PLuG to Deformable DETR**  As shown in Table 3, PLuG consistently improves performance across diverse Deformable DETR (Zhu et al., 2020) configurations. In single-scale experiments where only a single feature level is employed, the parameter overhead introduced by PLuG is minimal ($\sim 0.07\%$). Despite this negligible increase, we observe a slight but consistent performance gain from 39.4 AP to 39.7 AP and 41.4 AP to 41.7 AP, demonstrating the effectiveness of PLuG even in the simplest configuration. Remarkably, PLuG also demonstrates effectiveness when applied exclusively within the cross-attention modules of the decoder. Specifically, when PLuG is applied only to the decoder, AP improves from 44.4 to 44.7, indicating that the gating mechanism not only benefits encoder self-attention and decoder cross-attention layers but can also effectively modulate object queries.

Table 3: Results of applying PLuG to Deformable DETR (Zhu et al., 2020). [†] denotes our re-implementation and [*] indicates tests performed under mmdetection (Chen et al., 2019) framework in our setting.

| Model | AP | AP$_{50}$ | AP$_{75}$ | AP$_S$ | AP$_M$ | AP$_L$ | #Params | FLOPs | Inference FPS |
|---|---|---|---|---|---|---|---|---|---|
| Deformable DETR (single scale)[†] (Zhu et al., 2020) | 39.4 | 60.1 | 41.9 | 19.8 | 43.6 | 56.2 | 33.8M | 78G | 42.6* |
| + **PLuG** | **39.7** | 60.1 | 42.3 | 20.1 | 43.5 | 56.2 | 33.8M | 78G | 41.5* |
| Deformable DETR (single scale, DC5)[†] (Zhu et al., 2020) | 41.4 | 61.6 | 44.6 | 23.4 | 45.2 | 56.9 | 33.8M | 128G | 37.4* |
| + **PLuG** | **41.7** | 61.7 | 44.9 | 23.3 | 45.6 | 57.0 | 33.8M | 129G | 36.4* |
| Deformable DETR[†] (Zhu et al., 2020) | 44.4 | 63.2 | 48.5 | 25.7 | 47.8 | 59.4 | 39.8M | 173G | 27.4* |
| + **PLuG** (encoder only) | 44.7 | 63.7 | 48.5 | 26.6 | 48.3 | 59.4 | 39.8M | 173G | - |
| + **PLuG** (decoder only) | 44.7 | 63.4 | 48.4 | 27.0 | 48.0 | 59.4 | 39.8M | 173G | - |
| + **PLuG** | **44.9** | 63.7 | 48.8 | 26.9 | 48.3 | 59.3 | 39.9M | 174G | 26.8* |
| Iterative bounding box refinement[†] (Zhu et al., 2020) | 45.7 | 64.4 | 49.4 | 27.7 | 48.9 | 61.7 | 40.6M | 173G | 25.6* |
| + **PLuG** | **46.1** | 64.8 | 49.8 | 27.9 | 49.4 | 62.1 | 40.7M | 174G | 25.6* |
| Two-stage Deformable DETR[†] (Zhu et al., 2020) | 46.8 | 65.8 | 50.6 | 29.3 | 49.9 | 61.9 | 40.9M | 173G | 25.4* |
| + **PLuG** (encoder only) | **47.0** | 65.8 | 50.9 | 29.0 | 50.1 | 62.4 | 41.0M | 174G | 25.4* |

Furthermore, in the full Deformable DETR and the variant with iterative bounding box refinement, PLuG achieves 0.4 AP gain with only a slight increase in parameters ($\sim 0.24\%$), underscoring its efficiency and effectiveness. In the two-stage Deformable DETR configuration, however, applying PLuG to both the encoder and decoder leads to a slight performance drop, likely due to reduced proposal diversity. Therefore, we restrict PLuG to the encoder in this setting, which still yields an improvement from 46.8 to 47.0 AP with minimal parameter overhead. A marginal decrease in inference speed (FPS) is observed due to additional gating operations in single-scale setting, but as model complexity increases, the FPS difference introduced by PLuG becomes insignificant. Since multi-scale deformable attention is a key component in various vision architectures, the consistent improvements indicate strong potential for extending PLuG to a broad range of vision tasks and modalities.

## 4.2 ABLATION STUDIES

We conduct ablation studies to examine the effects of key design choices: (1) varying the gating-dimension fraction, (2) comparing Gating Modulation Layer (GML) variants, (3) assessing the activation function, and (4) evaluating head-specific versus shared gating. All ablations are conducted on DeiT (Touvron et al., 2021a), a pure attention-based architecture that enables clear analysis of each component's impact.

**Ablation on Gating Dimension Fraction** Figure 4 shows an exploration the gating dimension fraction in PLuG on DeiT-Ti (Touvron et al., 2021a), varying it from 0.125 to 2.0 relative to each head's subspace. Even small fractions (0.125–0.5) already beat the 72.2% baseline—72.48%, 72.46%, and 72.60%—with only 5.75–5.86M parameters. Increasing to 1.0 yields the best result, 73.2% at 6.01M parameters. Pushing further to 2.0 inflates parameters

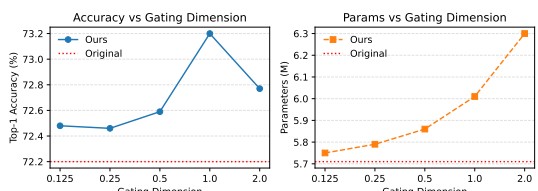

Figure 4: Ablation on gating dimension fraction. Accuracy and parameter count are shown.

to 6.3M but drops accuracy to 72.8%, indicating that exceeding the head dimension adds cost without benefit. Accordingly, we adopt a gating fraction of 1.0 for all experiments.

**Ablation on GML Variants** We perform an ablation study to assess the impact of Gating Modulation Layer (GML) variants in Table 4. The simplest variant (A), which excludes learnable modulation parameters, achieves 72.6% accuracy. Variant (B) adds two learned scalar factors in an additive manner and yields 72.8%, while variant (C) uses a single learned scalar and reaches 73.0%. Variant (D) incorporates a two-layer MLP with an intermediate hidden dimension ($h_{int} = 4$) but results in a slightly lower accuracy of 72.9%, possibly due to increased optimization complexity. Our proposed variant (E) combines two learned factors via elementwise multiplication and achieves the highest accuracy of 73.2%, demonstrating the effectiveness of pairwise multiplicative gating in modulating token-pair interactions.

Table 4: Ablation study of gating modulation layers (GML). Variant (E) performs best.

| Variant | Method | Top-1 Acc (%) |
|---|---|---|
| (A) | $G = \tanh(Q_{\text{gate}} K_{\text{gate}}^\top)$ | 72.6 |
| (B) | $G = \tanh(g_{\text{A}} + g_{\text{B}})$ | 72.8 |
| (C) | $G = \tanh(g),\ W_\phi \in \mathbb{R}^{1 \times 1}$ | 73.0 |
| (D) | $W_\phi^{(1)} \in \mathbb{R}^{1 \times h_{\text{int}}},\ W_\phi^{(2)} \in \mathbb{R}^{h_{\text{int}} \times 2}$ | 72.9 |
| (E) | $G = \tanh(g_{\text{A}} \odot g_{\text{B}})$ | **73.2** |

Table 5: Effect of activation functions in PLuG applied to DeiT (Touvron et al., 2021a).

| Model | Function | Top-1 Acc (%) |
|---|---|---|
| DeiT-Ti (Touvron et al., 2021a) + **PLuG** | tanh | 73.2 |
| DeiT-Ti (Touvron et al., 2021a) + **PLuG** | hardtanh | 71.0 |
| DeiT-Ti (Touvron et al., 2021a) + **PLuG** | softsign | 72.8 |
| DeiT-Ti (Touvron et al., 2021a) + **PLuG** | w/o tanh | 73.2 |
| DeiT-S (Touvron et al., 2021a) + **PLuG** | w/o tanh | ✗ |

**Effect of Activation Functions**  The influence of the activation function in the PLuG is summarized in Table 5. Hardtanh and softsign were included in the ablation as they behave similarly to tanh by providing bounded, symmetric nonlinear mappings. On DeiT-Ti (Touvron et al., 2021a), hardtanh reduced accuracy to 71.0%, softsign improved results to 72.8%, and tanh achieved 73.2%. Removing the nonlinearity entirely also reached 73.2% on DeiT-Ti, but training on DeiT-S became unstable and the loss diverged to NaN. These findings indicate that while bounded nonlinearity helps stabilize optimization, tanh provides the best trade-off between expressiveness and robustness.

Table 6: Comparison of head-specific gating in terms of accuracy, parameter count, and FLOPs. **h.s** denotes head-specific gating.

| Method | Top-1 Acc (%) | #Params | FLOPs |
|---|---|---|---|
| DeiT-Ti (Touvron et al., 2021a) + **PLuG** w/o **h.s** | 73.2 | 6.0M | 1.4G |
| DeiT-Ti (Touvron et al., 2021a) + **PLuG** w/ **h.s** | 73.1 | 6.6M | 1.5G |
| DeiT-S (Touvron et al., 2021a) + **PLuG** w/o **h.s** | 80.3 | 22.6M | 4.9G |
| DeiT-S (Touvron et al., 2021a) + **PLuG** w/ **h.s** | 80.1 | 25.5M | 5.5G |

**Effect of Head-Specific Gating**  We further examine the effect of applying gating coefficient matrix $G$ independently to each attention head (head-specific) versus sharing across all heads (our default). As shown in Table 6, head-specific gating on DeiT-Ti (Touvron et al., 2021a) increases the parameter count from 6.0M to 6.6M while slightly reducing accuracy from 73.2% to 73.1%. For the larger DeiT-S model, head-specific gating increases the parameter count increases from 22.6M to 25.5M and yields a small drop in accuracy from 80.3% to 80.1%. In addition, head-specific gating leads to a non-trivial increase in computational cost, with FLOPs rising from 1.4G to 1.5G on DeiT-T and from 4.9G to 5.5G on DeiT-S. These results indicate that head-shared gating is more efficient in both parameter count and computational cost, while remaining sufficient for capturing effective token-pair interactions.

### 4.3 FURTHER ANALYSIS

**Depthwise Analysis of** $G$  We investigate the gating behavior of PLuG across different transformer layers by analyzing the average $G$ values within DeiT-Ti and DeiT-S (Touvron et al., 2021a) in Figure 5. $G$ values are computed using images across all ImageNet-1K (Russakovsky et al., 2015) validation set. Across both architectures, we observe a consistent layerwise gating pattern: strongly positive gate values in the initial layer, near-zero values in the middle, and increasingly negative values in deeper layers. This progressive shift from amplification to sup-

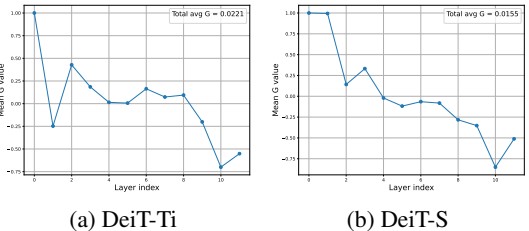

(a) DeiT-Ti          (b) DeiT-S

Figure 5: Layerwise mean $G$ values for (a) DeiT-Ti and (b) DeiT-S (Touvron et al., 2021a).

pression of token interactions emerges consistently across different model capacities. Interestingly, despite notable variability across layers, the mean $G$ value across layers remain near zero (DeiT-Ti: 0.0221, DeiT-S: 0.0155). These results suggest that PLuG operates as a depth-aware gating mechanism that adaptively reallocates attention strength across token pairs in a balanced manner, rather than applying uniform scaling across layers of the model.

**Attention-Pattern Similarity** On DeiT-S, we measure Centered Kernel Alignment (CKA) (Kornblith et al., 2019) between per-layer attention patterns, where the class token distributes attention over image regions, using the full ImageNet-1k validation set. Figure 6 shows that applying PLuG reduces similarity across the middle layers (3–8), reflecting lower redundancy and more gradual attention changes, while similarity increases in the upper layers (8–12), indicating smaller adjustments closer to the output. These patterns emerge naturally during training and highlight how PLuG reshapes attention across depth.

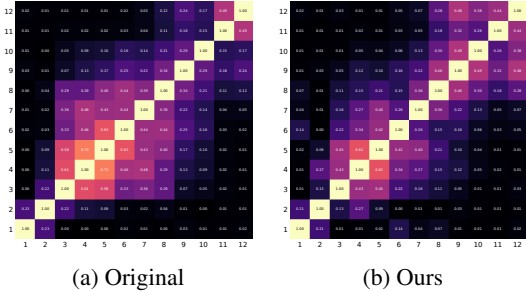

(a) Original      (b) Ours

Figure 6: CKA (Kornblith et al., 2019) between attention patterns of DeiT-S (Touvron et al., 2021a): (a) original and (b) PLuG.

## 4.4 VISUALIZATION

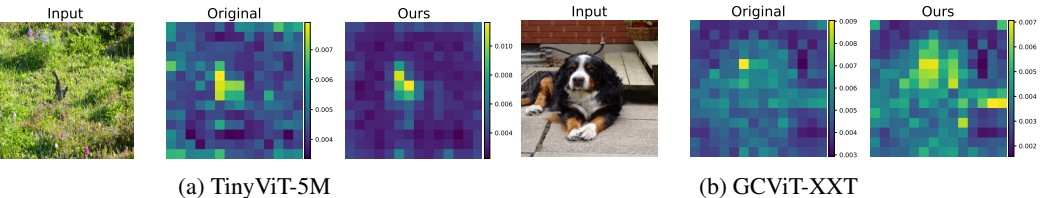

(a) TinyViT-5M      (b) GCViT-XXT

Figure 7: Attention rollout (Abnar & Zuidema, 2020) maps of (a) TinyViT-5M (Wu et al., 2022) and (b) GCViT-XXT (Hatamizadeh et al., 2023). Colour bars indicate relative attention strength.

We visualize the impact of PLuG using attention rollout (Abnar & Zuidema, 2020) maps, comparing against a vanilla TinyViT-5M (Wu et al., 2022) and GCViT-XXT (Hatamizadeh et al., 2023). The rollout aggregates attention weights across transformer layers, revealing how effectively the model attends from the class token to image patches. Figure 7 shows that PLuG consistently guides the model to focus on semantically informative regions (e.g., foreground objects) while suppressing spurious background features, highlighting PLuG's ability to enhance interpretability and robustness in vision transformers.

## 5 LIMITATIONS AND FUTURE WORK

Although PLuG consistently improves accuracy, the performance gain becomes less substantial as model capacity increases. As described in Section 4, smaller backbones benefit more noticeably, while larger models see more modest improvements, suggesting that the impact of token-pair gating is reduced when the base model is already highly expressive. In addition, the introduction of gating projections inevitably increases parameters, FLOPs, and memory usage. Future work will explore strategies to reduce these overheads and enhance efficiency, particularly for high-capacity models where the performance gains are limited.

## 6 CONCLUSION

In this paper, we present PLuG attention, a simple yet effective plug-in enhancement to self-attention that introduces learnable token-pair-specific gates to modulate attention logits prior to softmax operation. This mechanism selectively amplifies informative interactions and suppresses spurious ones, yielding more expressive and discriminative representations. PLuG integrates seamlessly into diverse attention variants and architectures, applied only within the attention module, with no architectural redesign and no hyperparameter tuning required. Across image classification, detection, and segmentation task, PLuG achieves consistent gains, demonstrating a broadly applicable and scalable enhancement for attention-based vision models.

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

# A PSEUDOCODE

---
**Algorithm 1** Pseudocode for `PLuG-Attention`.

---

```
# X: [B, N, C] tokens
# h: number of heads; d = C / h

def PLuGAttention(X, W_qkv, W_g):
    # main & gate projections
    Q, K, V = split(X @ W_qkv, 3)
    Qg, Kg  = split(X @ W_g, 2)

    s = 1 / sqrt(d)

    # main logits
    A = (Q @ K.transpose(-1, -2)) * s

    # raw gating logits
    R = (Qg @ Kg.transpose(-1, -2)) * s

    # GML
    gA, gB = split(Linear(R), 2)
    G = tanh(gA * gB)

    # apply and normalize
    W = softmax(A * (1 + G), dim=-1)
    O = W @ V

    return O
```

---

Algorithm 1 presents the PyTorch-style (Paszke et al., 2019) pseudocode for PLuG attention. The formulation allows seamless integration into existing architectures with minimal modification, supporting both experimental research development and efficient deployment.

# B BASELINE ARCHITECTURES

## B.1 VISION TRANSFORMER

The Vision Transformer (ViT) (Dosovitskiy et al., 2021) partitions an input image into non-overlapping patches, linearly projects each patch into an embedding, and prepends a learnable class token to form the input sequence. This sequence is processed by an encoder composed of stacked Transformer blocks (Vaswani et al., 2017), each containing a multi-head self-attention (MHSA) layer and a position-wise feed-forward network (FFN), with residual connections (He et al., 2016) and layer normalization (Ba et al., 2016). Positional encodings are added to retain spatial information, enabling ViT to operate directly on patch tokens without convolutional inductive biases.

## B.2 DEFORMABLE DETR.

DETR (Carion et al., 2020) is composed of a backbone, an encoder, and a decoder. The backbone (He et al., 2016) extracts feature map and the encoder applies stacked layers of multi-head self-attention (MHSA) and feed-forward networks (FFN) to refine these features. The decoder applies layers of cross-attention, MHSA, and FFNs to iteratively update object queries and predict class labels and bounding boxes. During training, each query is supervised to correspond to either a foreground object or background. Deformable DETR (Zhu et al., 2020) replaces the MHSA layers in the encoder and the cross-attention modules in the decoder with multi-scale deformable attention for improved efficiency and multi-scale feature aggregation.

## B.3 Mask2Former

Mask2Former (Cheng et al., 2022) is a universal architecture for image segmentation tasks, including semantic, instance, and panoptic segmentation. It employs a hierarchical backbone (He et al., 2016; Liu et al., 2021) to extract multi-scale features, which are encoded by a pixel decoder into a shared feature representation. A transformer decoder processes a fixed set of learnable queries through layers of cross-attention, self-attention, and feed-forward networks, iteratively predicting segmentation masks. By decoupling mask prediction from class prediction, Mask2Former achieves strong generalization across diverse segmentation tasks.

## C Experimental settings

### C.1 Datasets

**Image Classification.** ImageNet-1K (Russakovsky et al., 2015) is the standard large-scale benchmark for image classification, containing 1.2M training images and 50K validation images across 1,000 object categories. Its broad diversity of object categories makes ImageNet-1K a strong benchmark for evaluating the generalization of visual backbones.

**Object Detection.** COCO 2017 (Lin et al., 2014) is widely adopted for object detection, consisting of 118K training and 5K validation images annotated with bounding boxes for 80 categories. Images depict complex natural scenes with multiple instances and large-scale variation, posing significant challenges for localization and detection.

**Semantic Segmentation.** ADE20K (Zhou et al., 2019) provides a diverse benchmark for semantic segmentation, with 20K training and 2K validation images labeled at the pixel level across 150 semantic categories. Its indoor-outdoor scenes and fine-grained boundaries require models to leverage both global context and local detail.

### C.2 Implementation details

**ViT-based Architectures** We evaluate PLuG on several vision transformer models, including DeiT (Touvron et al., 2021a), TinyViT (Wu et al., 2022), Visformer (Chen et al., 2021), LV-ViT (Yuan et al., 2021), XCiT (El-Nouby et al., 2021), PVT (Wang et al., 2021), GV ViT (Hatamizadeh et al., 2023), ViTAE (Xu et al., 2021), and T2T-ViT (Yuan et al., 2021). For XCiT, we apply PLuG within the cross-covariance attention (XCA) layer. We follow the training configurations from the original papers or official repositories and all experiments were conducted using 4 NVIDIA RTX 3090 GPUs. For DeiT-S and GCViT, we use the training setups from the timm[1] implementation.

**Deformable DETR** We apply PLuG to the multi-scale deformable attention modules in Deformable DETR (Zhu et al., 2020). To reproduce performance close to the official github[2] implementation in our GPU setting, we use a batch size of 4 with gradient accumulation over 8 steps and reduce the weight decay to 5e-5. All other training settings follow the original configuration and PLuG is applied under this reimplementation setup across all evaluated variants of Deformable DETR. Training was performed on two NVIDIA RTX 4090 GPUs. Each configuration was run three times, and we report the mean over the three runs. Inference speed was measured on a single NVIDIA RTX 4090 GPU.

**Mask2Former** We insert PLuG only into the transformer decoder's MHSA layers, leaving the pixel decoder's multi-scale deformable attention and the decoder's masked cross-attention unchanged. No additional hyper-parameter tuning is applied beyond the default Mask2Former (Cheng et al., 2022) settings. The experiments were performed on 4 NVIDIA RTX 3090 GPUs.

---

[1]https://github.com/huggingface/pytorch-image-models
[2]https://github.com/fundamentalvision/Deformable-DETR

# D COMPLEXITY ANALYSIS

## D.1 COMPLEXITY OF STANDARD ATTENTION

In a Vision Transformer (Dosovitskiy et al., 2021), the forward cost comprises projections and attention. Forming $Q_{\text{main}}, K_{\text{main}}, V$ costs $3NC^2$, and the output projection adds $NC^2$. Computing scores $Q_{\text{main}} K_{\text{main}}^{\top}$ costs $HN^2 d = N^2 C$, and applying weights to values costs another $N^2 C$. Softmax/dropout over the $N \times N$ scores are lower-order $O(HN^2)$. The leading compute is

$$4NC^2 + 2N^2 C. \tag{13}$$

When $N \gtrsim 2C$, the $N^2 C$ terms dominate, otherwise the $NC^2$ projections dominate. Activations are $O(NC)$ for input and output, $O(3NC)$ for $Q, K, V$, and $O(HN^2)$ for attention maps.

## D.2 COMPLEXITY OF PLuG ATTENTION

PLuG augments MHSA with a head-shared gating branch. Main projections cost $4NC^2$ in total, and gate projections $x \to (Q_{\text{gate}}, K_{\text{gate}}) \in \mathbb{R}^{N \times d_g}$ cost $2NC d_g$. Main scores and application to $V$ cost $2N^2 C$, and the head-shared gated score $Q_{\text{gate}} K_{\text{gate}}^{\top}$ costs $N^2 d_g$. The Gating Modulation Layer (GML) contributes $O(N^2)$ elementwise work, and logit modulation adds $O(HN^2)$. Summing leading terms:

$$4NC^2 + \left(2N^2 C + N^2 d_g\right) + \underbrace{2NC d_g}_{\text{gate projection}} + \underbrace{O(N^2)}_{\text{GML}} + \underbrace{O(HN^2)}_{\text{logit modulation}}. \tag{14}$$

No new $C^2$ terms beyond projections are introduced. Activations remain $O(NC)$ for input and output, $O(3NC)$ $(Q, K, V)$, and $O(HN^2)$ (per-head maps), plus $O(Nd_g)$ for $(Q_{\text{gate}}, K_{\text{gate}})$ and one head-shared $O(N^2)$ gate matrix.

## D.3 COMPLEXITY OF MSDEFORMATTN

Let $C$ be channels, $H$ heads, $L$ feature levels, $P$ sampling points per head–level, $N_q$ queries, and $N_k = \sum_{\ell=1}^{L} S_\ell$ total key–value tokens. The value projection over keys costs $N_k C^2$, the output projection costs $N_q C^2$. Query linears for offsets/logits map $C \to HLP$ and cost $3N_q CHLP$. Bilinear sampling and aggregation over $LP$ points contribute $5N_q LP C$. The leading cost is

$$N_k C^2 + N_q C^2 + 3N_q CHLP + 5N_q LP C. \tag{15}$$

There is no $N_q N_k$ attention matrix, so cost scales linearly with spatial tokens.

## D.4 COMPLEXITY OF PLuG MSDEFORMATTN

PLuG adds a head-specific, level-specific (point-shared) gate on the query side. Relative to equation 15, the additional costs are

$$\underbrace{N_q CHL}_{\text{gate projection}} + \underbrace{O(N_q HL)}_{\text{GML}} + \underbrace{O(N_q HLP)}_{\text{logit modulation}}. \tag{16}$$

Thus the total leading cost is equation 15 + equation 16. No new $C^2$ terms are introduced. For large $C$ and modest $H, L, P$, the $C^2$ terms dominate, and PLuG overhead is small. Activations match MSDeformAttn order, with one additional $O(N_q HL)$ gate tensor.

# E   FURTHER ABLATIONS

## E.1   MORE DETAILS OF PLuG TO MULTI-SCALE DEFORMABLE ATTENTION

Table 7: Results of applying PLuG to the decoder's self-attention layer. [†] denotes our re-implementation.

| Model | Method | AP |
|---|---|---|
| Deformable DETR[†] (Zhu et al., 2020) | Vanilla | 44.5 |
| Deformable DETR (Zhu et al., 2020) + **PLuG** | Decoder self-attn | 44.2 |

**PLuG to Decoder Self-Attention**   As reported in Table 3, inserting PLuG into the encoder's multi-scale deformable attention and the decoder's cross-attention layers leads to consistent performance improvements. In contrast, applying PLuG to the decoder's self-attention layer results in a slight drop in AP, as shown in Table 7, suggesting that logit-level gating is less effective for self-attention refinement than for enhancing cross-attention or multi-scale aggregation in Deformable DETR (Zhu et al., 2020).

Table 8: Effect of applying tanh to $G$ in multi-scale deformable attention. [†] denotes our re-implementation

| Model | Method | AP |
|---|---|---|
| Deformable DETR[†] (Zhu et al., 2020) | Vanilla | 44.5 |
| Deformable DETR (Zhu et al., 2020) + **PLuG** | w/o tanh | 44.9 |
| Deformable DETR (Zhu et al., 2020) + **PLuG** | w/ tanh | 44.2 |

**Effect of Nonlinear Activation**   Experiments on ViT-based architectures reported in Table 5 indicate that bounded activations such as tanh or softsign yield stable or slightly improved accuracy. On the other hand, introducing a tanh nonlinearity to $G$ in multi-scale deformable attention reduces AP to 44.2, as shown in Table 8. These findings suggest that saturation functions help regularize dense query–key interactions in ViT (Dosovitskiy et al., 2021), but disrupt the fine-grained head–level gating signals in Deformable DETR, thereby diminishing its effectiveness.

Table 9: Ablation of gating granularity. Per-head and per-level gating with point sharing performs best.

| Model | Gating granularity | AP |
|---|---|---|
| | Head-specific, Level-specific, Point-shared (default) | **44.9** |
| | Head-specific, Level-specific, Point-specific | 44.8 |
| Deformable DETR (Zhu et al., 2020) + **PLuG** | Head-specific, Level-shared, Point-shared | 44.5 |
| | Head-shared, Level-specific, Point-specific | 44.5 |
| | Head-shared, Level-specific, Point-shared | 44.5 |

**Gating granularity of MSDeformAttn**   We vary the gating granularity of MSDeformAttn across heads, levels, and sampling points in Table 9. The default head-specific, level-specific, point-shared design achieves the best result with 44.9 AP. Making the gate point-specific offers no benefit, yielding 44.8 AP. A similar pattern is shown in Table 6, where head-specific gating in DeiT (Touvron et al., 2021a) models increases parameters and FLOPs without improving accuracy. Sharing across heads or across levels consistently reduces accuracy to 44.5 AP, showing that per-head and per-level specificity is important. These findings indicate that finer granularity is not always beneficial. Effective gating requires balancing expressiveness and efficiency, and we therefore adopt head-specific and level-specific gating with point sharing as the most effective choice.

## E.2   PRE-SOFTMAX VS. POST-SOFTMAX

We analyze the impact of gating position of $G$ within the PLuG attention mechanism on the DeiT-Ti (Touvron et al., 2021a). Specifically, we compare two gating strategies: pre-softmax gating (the

Table 10: Effect of gating position in PLuG on DeiT-Ti (Touvron et al., 2021a). Pre-softmax gating outperforms post-softmax gating.

| Model | Method | Top-1 Acc (%) |
|---|---|---|
| DeiT-Ti (Touvron et al., 2021a) | Vanilla | 72.2 |
| DeiT-Ti (Touvron et al., 2021a) + **PLuG** | Pre-softmax | 73.2 |
| DeiT-Ti (Touvron et al., 2021a) + **PLuG** | Post-softmax | 72.3 |

proposed approach) versus post-softmax gating. When $G$ is applied after softmax normalization, final attention logits $\tilde{A}_{\mathrm{main}}$ become:

$$\tilde{A}_{\mathrm{main}} = \mathrm{softmax}(A_{\mathrm{main}}) \odot (1 + G), \tag{17}$$

This strategy yields only a minor accuracy improvement from 72.2% to 72.3%, as can be seen in Table 10. The limited effectiveness of post-softmax gating can be attributed to two factors: (1) Post-softmax gating disrupts the probability distribution, causing attention weights to no longer sum to one and negatively impacting the learned attention balance. (2) Since softmax normalization compresses logits into a narrow probability range, gating applied afterward has limited ability to substantially adjust attention distribution patterns. This ablation underscores the importance of applying $G$ at the logit level (pre-softmax), enabling more expressive and beneficial modulation of attention scores, and supports logit-level gating as the superior design choice within the PLuG framework.

### E.3 ROLE OF RESIDUAL GATING.

Table 11: Effect of residual gating in PLuG applied to DeiT (Touvron et al., 2021a).

| Model | Method | Top-1 Acc (%) |
|---|---|---|
| DeiT-Ti (Touvron et al., 2021a) + **PLuG** | w/ residual | 73.2 |
| DeiT-Ti (Touvron et al., 2021a) + **PLuG** | w/o residual | 68.9 |

Table 12: Effect of residual gating in PLuG applied to Deformable DETR (Zhu et al., 2020).

| Model | Method | AP |
|---|---|---|
| Deformable DETR (Zhu et al., 2020) + **PLuG** | w/ residual | 44.9 |
| Deformable DETR (Zhu et al., 2020) + **PLuG** | w/o residual | 43.3 |

Table 11 and Table 12 present the effect of incorporating a residual connection in the gating pathway. For DeiT-Ti (Touvron et al., 2021a), removing the residual connection leads to a clear drop in Top-1 accuracy from 73.2% to 68.9%. For Deformable DETR (Zhu et al., 2020), AP decreases from 44.9 to 43.3 without residual gating. These results indicate that residual gating is essential for stable optimization and reliable accuracy across both dense and sparse attention mechanisms within PLuG.

# F COMPARISON WITH OTHER METHODS

In this section, we compare PLuG against prior attention variants, including those covered in Section 2 as well as more recent approaches, using four simple criteria.

- **Pre-softmax**: Does the method explicitly modify attention scores before the softmax?

- **Granularity**: Is it modulating individual query–key scores, either amplifying or suppressing them?

- **Generalizability**: Has it shown gains across multiple architectures and tasks in its demonstrated domain?

- **Scalability**: Can it seamlessly integrated into various attention variants?

Table 13: Comparison of attention variants under four criteria. ✓ indicates the criterion is satisfied, while ✗ does not.

| Model | Pre-softmax | Granularity | Generalizability | Scalability |
|---|---|---|---|---|
| Talking Heads (Shazeer et al., 2020) | ✓ | ✓ | ✓ | ✗ |
| Synthesizer (Tay et al., 2021) | ✓ | ✗ | ✓ | ✗ |
| Gated Attention (Doering & Gall, 2023) | ✓ | ✗ | ✗ | ✗ |
| Selective Self-Attention (Zhang et al., 2024a) | ✓ | ✗ | ✓ | ✗ |
| Selective Attention (Leviathan et al., 2025) | ✓ | ✗ | ✓ | ✗ |
| Forgetting Transformer (Lin et al., 2025) | ✓ | ✗ | ✓ | ✓ |
| Structured Attention (Kuang et al., 2025) | ✓ | ✓ | ✓ | ✓ |
| **PLuG** (Ours) | ✓ | ✓ | ✓ | ✓ |

**Talking-Heads** Talking-Heads (Shazeer et al., 2020) augments attention by mixing across heads through small learned projections on the pre-softmax logits together with a light post-softmax mix, which improves inter-head coordination without requiring special hyperparameter tuning. PLuG takes a different approach by rescaling each logit with a learned, content-dependent factor $G$, enabling per-token-pair modulation directly in pre-softmax and avoiding any post-softmax step. Both methods are lightweight and can be used as plug-in replacements. Talking-Heads has been validated primarily in NLP (Lan et al., 2019; Devlin et al., 2019; Raffel et al., 2020), while PLuG shows consistent gains across vision tasks. Although Talking-Heads has also been applied to vision models such as CaiT (Touvron et al., 2021b) with improvements, it has not been demonstrated to integrate seamlessly with diverse attention variants as PLuG does.

**Synthesizer** Synthesizer (Tay et al., 2021) replaces the dot-product operation with either a *Dense* variant, where a token-wise MLP produces a row-wise attention map for each query, or a *Random* variant, where weights are sampled without content dependence. These designs show that competitive language modeling performance is possible even without explicit query–key similarity. PLuG takes a different approach by augmenting rather than discarding the dot product, applying a content-dependent gate to each logit to preserve pairwise structure while improving selectivity. Both operate in pre-softmax space, but Synthesizer produces full attention maps directly, whereas PLuG modulates logits at the per-token-pair level. While Synthesizer demonstrates generalizability across NLP benchmarks, it *cannot replace cross-attention*, whereas PLuG has been successfully applied to cross-attention in Deformable DETR (Zhu et al., 2020).

**Gated Attention** The Gated Attention Transformer (Doering & Gall, 2023) is designed for multi-person pose tracking. It forms association scores by mixing two pre-softmax affinity maps, appearance similarity and pose-based geometric consistency, via a global $\alpha - Gate$ that creates a weighted sum before softmax, which *yields coarse control rather than independent per-pair modulation*. The approach introduces additional components for pose encoding, matching, and track management and depends on task-level thresholds and heuristics, so it is not a drop-in attention replacement. In contrast, PLuG is not tied to any specific task and provides a general logit-level modulation mechanism that can be seamlessly applied across architectures and vision tasks.

**Selective Self-Attention**   Selective Self-Attention (SSA) (Zhang et al., 2024a) introduces token-aware temperatures on queries and values: the query temperature scales the entire row of the logit matrix for each query (sharpening or smoothing its distribution over all keys), while the value temperature modulates the aggregation path. PLuG instead adjusts individual query–key logits, enabling finer-grained control. SSA shows improvements in language modeling, is lightweight and hyperparameter-free, and is *mainly evaluated in causal language modeling, where each token attends only to previous tokens*. PLuG focuses on vision models and tasks such as classification, detection, and segmentation, and is not tailored to causal language modeling.

**Selective Attention**   Selective Attention (Leviathan et al., 2025) derives a parameter-free, non-negative mask from an existing attention head and subtracts it from the pre-softmax logits, selectively suppressing query–key scores but never amplifying them. The method preserves the architecture and introduces no hyperparameters. In contrast, PLuG performs token-pair-wise logit rescaling that enables both amplification and suppression for finer control. Empirically, Selective Attention reports consistent gains in decoder-only language models and small improvements in the T5 decoder (Raffel et al., 2020), while also providing strong KV-cache pruning benefits. However, current evidence is limited to standard dense attention (Vaswani et al., 2017) rather than multi-query (Shazeer, 2019) or grouped-query (Ainslie et al., 2023) variants.

**Forgetting Transformer**   FoX (Lin et al., 2025) augments softmax attention with a data-dependent forget gate that adds a pairwise bias to the query–key scores before softmax. In terms of granularity, the method computes a bias for each query–key pair. Crucially, this bias is defined as the log of a product of forget factors, and because each factor is at most one, the resulting log term is non-positive. Thus, FoX can decrease a pair's score or leave it unchanged, but it cannot increase it. The paper reports consistent gains within NLP, including long-context language modeling, length extrapolation, and several downstream evaluations. Regarding scalability, FoX is a light change compatible with standard causal attention and FlashAttention (Dao, 2023).

**Structured Attention**   This approach replaces the standard dot-product scorer with structured matrices (MLR and BTT) that impose inductive biases and directly modify pre-softmax scores. The structured scorer produces a value for each query–key pair, enabling both increases and decreases at the pairwise level. Reported gains include in-context regression, language modeling with distance-aware compute, and long-range time-series forecasting, using standard Transformer backbones and a Chronos (Ansari et al., 2024) variant. For scalability, the method uses batched matrix multiplications and remains compatible with grouped-query attention and RoPE Su et al. (2024). However, unlike PLuG, *it introduces structural hyperparameters such as ranks, levels, and block sizes, which typically require design-specific tuning*.

# G  FURTHER VISUALIZATIONS

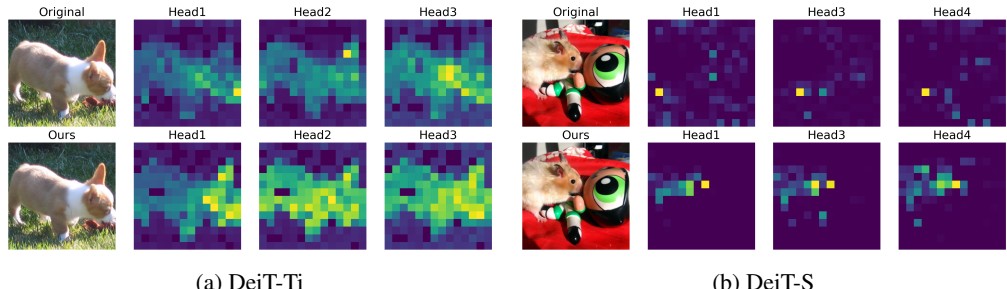

(a) DeiT-Ti                    (b) DeiT-S

Figure 8: Comparison of attention maps from the class token of DeiT-Ti (Touvron et al., 2021a). Brighter regions indicate stronger attention.

**Per-Head Attention Map**    To qualitatively illustrate the effect of the proposed PLuG mechanism, we visualize per-head attention maps for both the baseline DeiT (Touvron et al., 2021a) and the DeiT enhanced with PLuG. Specifically, we extract attention weights from the class token to all image regions in the final transformer block. For DeiT-S, we select representative heads that exhibit strong activations and present their corresponding maps. As shown in Figure 8, PLuG yields attention patterns that are more semantically concentrated than those of the baseline, suggesting that its improved focus on salient regions contributes to the observed performance gains.

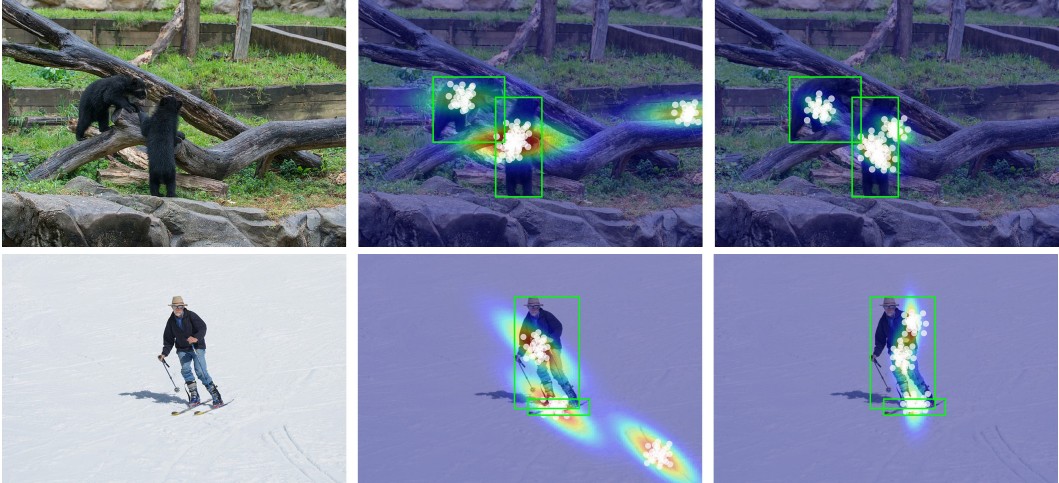

Figure 9: Qualitative comparison of sampling point distributions. Left: input image, middle: vanilla MSDeformAttn, right: PLuG-MSDeformAttn. Green boxes denote ground-truth object locations. White dots represent aggregated sampling points from the top-$k$ object queries at the highest-resolution feature level, and the jet-colored overlay shows their kernel density estimate.

**Sampling-Point Visualization**    We visualize sampling points from the final decoder layer of Deformable DETR (Zhu et al., 2020) to illustrate the effect of PLuG. We compare a baseline and a PLuG applied model that differ only in the multi-scale attention module. During inference, we record the sampling locations produced by the decoder's cross-attention. For each image, we select the top-$k$ object queries based on classification confidence, extract their sampling points from the highest-resolution feature map, and project them onto the resized input image. We overlay these points and their Gaussian kernel density estimates to qualitatively assess how PLuG alters the sampling distribution. As shown in Fig 9, PLuG places greater emphasis on informative regions while reducing sampling in irrelevant background areas.

