# OpenReview forum: "PLuG-Attention: Unleashing the Potential of Attention via Plug-in Pairwise Logit Gating"
_ICLR.cc/2026/Conference — ICLR 2026 Conference Withdrawn Submission_

### Official Review · Reviewer_CV4G · 2025-10-16

**Soundness:** 2
**Presentation:** 3
**Contribution:** 2
**Rating:** 4
**Confidence:** 4

**Summary:**

This paper introduces PLuG (Pairwise Logit Gating), a plug-in mechanism designed to enhance attention mechanisms in vision models by modulating attention logits at the token-pair level before softmax. Unlike standard attention that uniformly scores query–key interactions, PLuG introduces learnable gates that selectively amplify or suppress specific token-pair interactions. The authors demonstrate that PLuG can be seamlessly integrated into various architectures—including ViTs, Mask2Former, and Deformable DETR—without architectural redesign or hyperparameter tuning. Extensive experiments across classification, segmentation, and detection tasks show consistent performance improvements with slight computational overhead.

**Strengths:**

* This paper is well-organized and easy to follow.
* The method is validated across diverse architectures and tasks, consistent improvements are reported across multiple benchmarks (ImageNet-1K, ADE20K, COCO), with slight parameter and FLOP increases.

**Weaknesses:**

* Lack of insight. The paper focuses heavily on empirical validation but offers limited insight (e.g., theoretical grounding or intuitive explanation) for why logit-level gating works. Specifically,
  - Why cannot the attention logits itself learn the scoring and an extra gating module is necessary for that? How do you define "model capacity" (Sec. 5) and why the designed module increases it for, and only for small models?
  - Could the performance gain simply comes from extra parameters or FLOPs, instead of the concrete design of the gating logic?

* The design of Gating Modulation Layer (GML) is redundant and arbitrary. Sequentially applying linear projection, splitting and elementwise multiplication should be simply equivalent to a square activation function, why do you make it so complex?

**Questions:**

1. How does PLuG interact with other attention optimizations like FlashAttention or sparse attention—are there synergies or conflicts?
2. Could the authors elaborate on why PLuG performs poorly when applied to decoder self-attention (Table 7), and whether this limitation could be mitigated?

---

### Official Review · Reviewer_J3q1 · 2025-10-20

**Soundness:** 2
**Presentation:** 3
**Contribution:** 2
**Rating:** 4
**Confidence:** 4

**Summary:**

This paper proposes an enhanced attention mechanism called PLuG-Attention (Pairwise Logit Gating). By introducing a learnable gating mechanism, PLuG modulates the attention logits of every token pair prior to softmax normalization, enabling the model to selectively amplify informative interactions and suppress spurious interactions, thereby improving its ability to capture spatial and semantic relationships. The paper demonstrates performance improvements across various architectures and tasks.

**Strengths:**

1. PluG has a small number of parameters and computational complexity, and barely increases inference time.
2. PluG can simultaneously enhance and suppress interactions between tokens.
3. PluG is applicable to various attention mechanisms and tasks.

**Weaknesses:**

1. Although the paper compares with other methods in the appendix, it lacks quantitative results, which is very important for verifying the effectiveness of the method compared with previous work.
2. The method appears to achieve performance gains only on smaller-scale models. Since small models are inherently lightweight, the benefit of employing a lightweight attention enhancement mechanism on them is also limited.

**Questions:**

1. Have the authors attempted to conduct experiments on models with more than 200M parameters? I would like to know whether performance gains are still achieved when the model scale is larger.

2. I don't fully understand the mechanism behind the Gating Modulation Layer (GML). Why is the raw gating matrix projected into two factors that are then multiplied? Can the authors provide more intuition or explanation beyond just the ablation study results?

---

### Official Review · Reviewer_b9GB · 2025-10-29

**Soundness:** 1
**Presentation:** 3
**Contribution:** 1
**Rating:** 2
**Confidence:** 5

**Summary:**

This paper proposes PLuG Attention, a lightweight gating mechanism that modulates pre-softmax attention logits with a learned token-pair coefficient matrix. The design is simple, integrates seamlessly into ViTs, Mask2Former, and Deformable DETR, and achieves small but consistent improvements across classification, segmentation, and detection benchmarks.

**Strengths:**

1. Very clear and easy-to-implement mechanism.
2. Consistent improvements across multiple vision tasks.
3. Solid ablation studies and visual analyses that help explain the effect.
4. Pseudocode and mathematical description are easy to follow.

**Weaknesses:**

1. Novelty is limited. The method is closely related to existing gated attention variants and the conceptual overlap is strong. Importantly, the paper does not cite or compare against DGSA (Differential Gated Self-Attention) where already introduced input-dependent pairwise gating, but applied post-softmax to fuse excitatory/inhibitory maps. PLuG differs mainly in where the gating occurs and at the usage of single softmax branch. FoX introduces additive, input-dependent pre-softmax biases (suppressive only). PLuG’s multiplicative form can be seen as a proportional bias term AijGij. Overall, PLuG feels like another variant in a well-explored space rather than a fundamentally new mechanism.
2. Modest empirical gains, possibly within variance. The gains are relatively modest (+0.3–0.5% top-1 on ImageNet, +0.2–0.4 AP in DETR). The paper does report that each configuration was run three times and averaged. However, no standard deviations or confidence intervals are given. For small deltas (e.g. +0.1 mIoU), it is impossible to judge if they are meaningful. Reporting only the mean hides the possible overlap between baseline and PLuG runs. Improvements diminish with larger backbones.
3. No head-to-head comparisons with closest prior work. While Talking-Heads, ConVit, and FoX are mentioned, there are no direct baselines under the same settings. Without such comparisons, it is difficult to judge if PLuG is truly more effective than existing input-dependent gating approaches.
4. Unanalyzed failure cases. Applying PLuG to Deformable DETR decoder self-attention reduces AP, but no explanation is provided. This undermines the claim of “universality” and suggests limits that deserve deeper analysis.

**Questions:**

1. How do you formally distinguish PLuG from DGSA (Differential Gated Self-Attention), which also applies input-dependent pairwise gating (albeit post-softmax with excitatory/inhibitory maps)? Could you elaborate on what fundamentally new modeling capacity PLuG enables beyond simply moving the gate pre-softmax and apply on a single softmax attention channel?
2. Can you clarify whether your proportional bias view A'ij=Aij+AijGij is essentially equivalent to FoX with a logit-dependent bias? If not, what key expressive differences make PLuG more powerful?
3. How does PLuG compare empirically to DGSA, FoX, or ConVit in the same backbones? Since these are the most closely related mechanisms, empirical head-to-head evaluation would greatly clarify PLuG’s contribution.
Why does PLuG hurt performance in decoder self-attention? Can you provide an analysis of this failure (e.g., attention distribution sharpness, redundancy, or training instability)?
Given that performance gains diminish or vanish for larger backbones, do you see PLuG as mainly a small-model regularizer, or is there an intuition why scalability is limited?

---

### Official Review · Reviewer_8rjb · 2025-10-31

**Soundness:** 3
**Presentation:** 3
**Contribution:** 3
**Rating:** 4
**Confidence:** 3

**Summary:**

This paper proposes PLuG-Attention, a plug-in mechanism that introduces Pairwise Logit Gating to modulate attention logits before the softmax operation. The method learns a token-pair-specific gating coefficient matrix that selectively amplifies informative query–key interactions and suppresses spurious ones, aiming to enhance representational expressiveness and interpretability. PLuG is simple to implement, architecture-agnostic, and evaluated across a wide range of vision transformer architectures. Experiments show consistent performance improvements with minimal parameter and FLOP overhead.

**Strengths:**

1. Simple and Generalizable Design: PLuG can be easily integrated into existing architectures without redesign or tuning.

2. Broad Empirical Validation: Comprehensive experiments on classification, detection, and segmentation tasks confirm consistent gains.

3. Interpretability: Visualization and CKA analyses effectively demonstrate how PLuG modulates attention structure and focuses on semantically meaningful regions.

4. Clear written and visualization.

5. Practicality: The approach achieves improvements with negligible computational overhead, making it appealing for deployment.

**Weaknesses:**

1. Lack of Theoretical Foundation: The method is largely empirical, with limited theoretical justification for why pairwise gating enhances expressiveness or stability.

2. Limited Scaling and Generalization Analysis: The method is only tested on moderate-scale models; it’s unclear if gains persist at very large scales

3. Potential Redundancy: The added gating projections may overlap functionally with existing attention head mechanisms or MLP mixing.

**Questions:**

1. The baselines reported in Table 1 appear somewhat outdated; including comparisons with more recent vision backbones would strengthen the empirical evaluation.

2. It would be interesting to know whether PLuG can be applied to linear attention variants, such as PolaFormer [1] or FLatten Transformer [2], and if similar gains can be observed there.

3. The paper would benefit from additional analysis or intuition showing how the gating mechanism affects gradient flow or attention entropy, to better justify its stabilizing effect.

4. Finally, clarification is needed on whether PLuG introduces any training instability or noticeable memory overhead when applied to high-resolution or large-scale vision tasks.


This is an interesting work and I will raise my score if authors address all my concerns.

[1] Meng, W., Luo, Y., Li, X., Jiang, D. and Zhang, Z., 2025. PolaFormer: Polarity-aware linear attention for vision transformers. arXiv preprint arXiv:2501.15061.

[2] Han, Dongchen, et al. "Flatten transformer: Vision transformer using focused linear attention." Proceedings of the IEEE/CVF international conference on computer vision. 2023.

---

### Official Review · Reviewer_udxS · 2025-10-31

**Soundness:** 4
**Presentation:** 4
**Contribution:** 3
**Rating:** 8
**Confidence:** 4

**Summary:**

The paper proposes PLuG (Pairwise Logit Gating) attention, a simple plug-in mechanism that operates before the softmax and at the token-pair level, modulating the attention logits with a learnable gating matrix. The gate is produced from separate query/key projections (Q₍gate₎, K₍gate₎), passed through a lightweight Gating Modulation Layer (GML), and finally applied multiplicatively so that the original attention is preserved and only amplified/suppressed where needed. The method is dropped into several vision backbones (DeiT, TinyViT, XCiT, PVT, GCViT, Mask2Former, Deformable DETR) and consistently improves performance, especially on small/medium models, with very small parameter/FLOPs overhead. The paper is very well written, the related work is rich and up-to-date, and the qualitative visualizations make the idea easy to follow.

**Strengths:**

1. **Novel but simple idea.** Logit-level, token-pair gating is an under-explored spot between “talking heads” / row-wise temperatures and post-attention gating. Here it is done in the minimal way (extra Q/K, small GML, residual 1+G) and is easy to implement.
2. **Plug-in nature.** Authors actually show it on ViTs, Mask2Former, and Deformable DETR without surgery in the rest of the model, which supports the claim that it is a general mechanism and not tailored to one backbone.
3. **Clear writing + good figures.** The mechanism is explained in Section 3 with a clean decomposition (main path vs gating path vs GML) and the figures (Fig. 1–3, 6–7) really help.
4. **Solid experimental section.** ImageNet-1K classification across many small ViT-style models, ADE20K segmentation with Mask2Former, COCO detection with Deformable DETR, plus thorough ablations on gating dimension, GML variants, activations, head-specific vs shared gating. This is above average for this line of work.
5. **Honest limitations.** They themselves say gains are more pronounced on small models and become modest as capacity increases (Sec. 5). This matches the results tables.

**Weaknesses:**

1. **Effect mainly on small models.** This is already acknowledged, but it means the contribution is more about expressivity for small vision transformers than about scaling to very large models.
2. **Attention analysis is mostly qualitative.** Fig. 7 and the rollout maps are nice, and the CKA analysis (Fig. 6) shows reduced redundancy in mid-layers, but it stays at a descriptive level. This is exactly where one extra quantitative experiment would strengthen the story.
3. **Broadcasted gate across heads.** They justify why head-shared is better (Table 6), but this also means we don’t fully see whether PLuG creates complementary heads or just reshapes a shared pattern. A small quantitative probe could clarify that.

**Questions:**

- Right now, the paper gives qualitative attention maps (e.g. Fig. 7) but not a quantitative look at how PLuG changes attention diversity across heads. How complementary are the attention maps of the PLuG-augmented model compared to the original model? A simple way to do this is to borrow the complementarity / head-diversity metric from [1] and present the results.

[1] Psomas, Bill, et al. “Attention, Please! Revisiting Attentive Probing for Masked Image Modeling.” arXiv:2506.10178 (2025).

Moreover:
- Take a small model (e.g. DeiT-S with and without PLuG)
- For a fixed validation subset (ImageNet-1K val is fine), compute attention maps for all heads in selected layers
- Compute the complementarity / dissimilarity between heads within the baseline and within the PLuG model, and also the cross-model comparison (baseline head vs PLuG head).
- Report a small table showing whether PLuG increases head diversity or, alternatively, whether it re-organizes attention compared to the original model.

---

Overall recommendation: Accept

Justification: Method is novel yet simple, clearly written, broadly validated across tasks, and authors are transparent about limitations. A tiny quantitative attention experiment would polish the paper and provide some insights.

---

### Author Response · Authors · 2025-11-13

To all reviewers,

Thank you very much for your thoughtful evaluation and constructive feedback. I sincerely appreciate your careful comments and critical insights, which have helped me reflect deeply on the limitations of the current submission. Your detailed suggestions will be invaluable as I refine the idea and develop a stronger version of this work.

I am truly grateful for the opportunity to learn from your perspectives. Your feedback has provided an important motivation for my growth as a researcher. Thank you again for your time, effort, and insightful review.

---

### Note · Authors · 2025-11-13

I have read and agree with the venue's withdrawal policy on behalf of myself and my co-authors.